# Enhancing Ligand Validity and Affinity in Structure-Based Drug Design with Multi-Reward Optimization

Seungbeom Lee [* 1]   Munsun Jo [* 2]   Jungseul Ok [1 3]   Dongwoo Kim [1 3]

## Abstract

Deep learning-based structure-based drug design aims to generate ligand molecules with desirable properties for protein targets. While existing models have demonstrated competitive performance in generating ligand molecules, they primarily focus on learning the chemical distribution of training datasets, often lacking effective steerability to ensure the desired chemical quality of generated molecules. To address this issue, we propose a multi-reward optimization framework that fine-tunes generative models for attributes, such as binding affinity, validity, and drug-likeness, together. Specifically, we derive direct preference optimization for a Bayesian flow network, used as a backbone for molecule generation, and integrate a reward normalization scheme to adopt multiple objectives. Experimental results show that our method generates more realistic ligands than baseline models while achieving higher binding affinity, expanding the Pareto front empirically observed in previous studies.

## 1. Introduction

Designing ligand molecules that simultaneously optimize multiple objectives – such as binding affinity, synthetic accessibility, and strain energy – is a central challenge in *structure-based drug design* (SBDD). Conventional deep-learning approaches typically address only one or two objectives at a time, most commonly focusing on high binding affinity and basic chemical feasibility, e.g., through simple validity checks via the valency rule. While these methods have advanced the generative capabilities of molecular mod-

*Equal contribution [1]Graduate School of Artificial Intelligence, POSTECH, South Korea [2]KT Corporation, South Korea [3]Department of Computer Science & Engineering, POSTECH, South Korea. Correspondence to: Dongwoo Kim <dongwoo.kim@postech.ac.kr>.

*Proceedings of the $42^{nd}$ International Conference on Machine Learning*, Vancouver, Canada. PMLR 267, 2025. Copyright 2025 by the author(s).

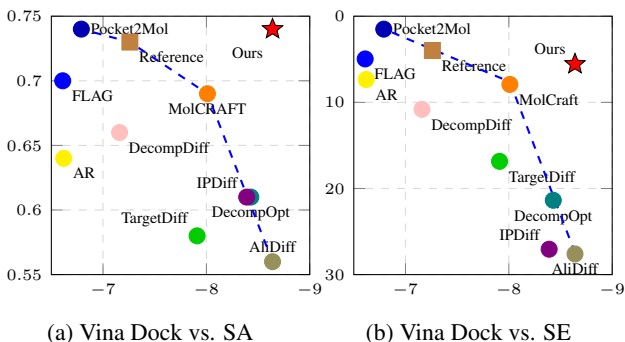

*Figure 1.* Comparison of the median Vina docking score (lower is better) against Synthetic accessibility (SA, higher is better) and Strain Energy (SE, lower is better). The dashed blue line represents a Pareto front obtained from previous works.

els (Huang et al., 2024; Qu et al., 2024; Guan et al., 2023), they often neglect other crucial factors like ease of conformation stability such as strain energy or synthesis. In practice, drug discovery demands a careful balance of many properties, making single-objective or narrowly focused solutions insufficient.

In current deep generative modeling for SBDD, standard strategies generally concentrate on minimizing the discrepancy between a generated ligand and a training set of protein-ligand complexes. Although this can lead to strong binding affinities, improving multiple objectives beyond binding, e.g., synthetic accessibility, usually requires careful tuning of loss terms or elaborate constraints. As a result, purely likelihood-based approaches can struggle to expand the *Pareto front*, which captures trade-offs among distinct objectives.

Figure 1 illustrates the trade-offs among Vina docking score measuring binding affinity to target protein, synthetic accessibility (SA), and strain energy (SE) using results from nine different generative models, along with a reference dataset included in the test set. Although recently proposed methods such as MolCRAFT (Qu et al., 2024), IPDiff (Huang et al., 2024), and AliDiff (Gu et al., 2024), outperform the reference data in terms of Vina docking score, they underperform on SA and SE. Ideally, a design process would push

the Pareto frontier toward both lower docking scores and higher SA (or lower SE). In contrast, these findings show that existing models remain overly focused on improving affinity, thus calling into question the practical utility of their generated molecules.

**Our Approach** Building on recent developments in preference-based reinforcement learning, we introduce a *multi-reward Direct Preference Optimization* (DPO) framework adapted for Bayesian Flow Networks (BFNs), which recently show promising performance on ligand generation tasks (Qu et al., 2024). DPO refines a pretrained generative model using feedback from external software or metrics, enabling fine-grained control over multiple properties (Rafailov et al., 2024; Kim et al., 2024). In our setting, we treat each property, such as binding affinity, synthetic accessibility, and strain energy, as a separate reward. We then apply softmax-based normalization to ensure commensurability of metrics with different ranges and adopt an uncertainty-regularized ensemble strategy to penalize high variance in multi-reward predictions. This multi-reward scheme expands the trade-off frontier between objectives, thereby improving molecular candidates across multiple dimensions.

## 2. Related Work

**Preference Alignment and Direct Preference Optimization (DPO)** Preference alignment has emerged as a powerful paradigm for adjusting pretrained models to better reflect desired behaviors. Originally adopted in large language models (LLMs) to align with human feedback (Ouyang et al., 2022), DPO (Rafailov et al., 2024) provides a more direct means of incorporating user or expert preferences compared to traditional reinforcement learning. Subsequently, DPO-based methods have been extended to text-to-image generative models (Wallace et al., 2023), showing that preference alignment is not limited to language domains. In multi-objective settings, Kim et al. (2024) propose a confidence-based or variance-penalized approach to combine heterogeneous reward signals, further enhancing stability and interpretability.

**Structure-Based Drug Design** Structure-based drug design (SBDD) focuses on generating or modifying small-molecule ligands to exhibit strong binding affinity for a target protein while maintaining favorable pharmacological properties. Classical computational approaches rely on de novo design or docking-based virtual screening (Schneider & Fechner, 2005) to identify candidate molecules, followed by iterative refinement and validation through experimental assays.

Recent deep learning methods cast SBDD as a conditional generative modeling task, leveraging protein-ligand complex data to learn a distribution over valid 3D conformations. Autoregressive approaches place atoms or fragments within protein pockets in a sequential manner (Luo et al., 2021; Peng et al., 2022), while diffusion models generate candidate molecules by denoising from random noise, learning continuous atom positions and discrete atom types jointly (Guan et al., 2023; Huang et al., 2024; Guan et al., 2024; Zhou et al., 2024). Most recently, Qu et al. (2024) introduced MolCRAFT, which uses Bayesian Flow Networks (BFNs) (Graves et al., 2023) to sample in parameter space rather than directly in data space, leading to improved binding affinity and more stable 3D molecular structures. To further improve the quality of binding affinity, Gu et al. (2024) introduced AliDiff, which fine-tunes a diffusion-based generative model via DPO.

On the other hand, Harris et al. (2023) raises questions about the evaluation practices with the deep generative models for SBDD. They find that many generated molecules from recent generative models violate physical constraints and call for an expanded evaluation of SBDD. A similar observation is made for deep learning based binding affinity prediction in Buttenschoen et al. (2024). Our approach builds upon this foundation by adopting a multi-reward perspective that jointly optimizes for binding affinity *and* other molecular properties (e.g., synthetic accessibility, strain energy), thereby expanding the Pareto frontier in multi-objective SBDD.

## 3. Preliminaries

We first introduce the problem formulation of structure-based drug design with notations used throughout this manuscript. We then provide an overview of Bayesian Flow Networks (BFNs)(Graves et al., 2023) and MolCRAFT(Qu et al., 2024), where the latter leverages BFNs to generate target molecules.

### 3.1. Problem Formulation

In structure-based drug design (SBDD), the goal is to generate ligand molecules conditioned on a given protein target. We denote a protein target by $\mathcal{P} = \{(\mathbf{x}_{\mathcal{P}}^{(i)}, \mathbf{v}_{\mathcal{P}}^{(i)})\}_{i=1}^{N_{\mathcal{P}}}$, where $\mathbf{x}_{\mathcal{P}}^{(i)} \in \mathbb{R}^3$ is the three-dimensional coordinate of the $i$-th atom, and $\mathbf{v}_{\mathcal{P}}^{(i)} \in \mathbb{R}^{D_{\mathcal{P}}}$ is a one-hot vector encoding the atom type. A molecule is analogously represented as $\mathcal{M} = \{(\mathbf{x}_{\mathcal{M}}^{(i)}, \mathbf{v}_{\mathcal{M}}^{(i)})\}_{i=1}^{N_{\mathcal{M}}}$, where $\mathbf{x}_{\mathcal{M}}^{(i)} \in \mathbb{R}^3$ and $\mathbf{v}_{\mathcal{M}}^{(i)} \in \mathbb{R}^{D_{\mathcal{M}}}$ denote the three-dimensional coordinate and the atom-type vector, respectively. For convenience, we denote each molecule by $\mathbf{m} = [\mathbf{x}, \mathbf{v}]$ where $[\cdot, \cdot]$ is the concatenation of $\mathbf{x} \in \mathbb{R}^{N_{\mathcal{M}} \times 3}$ and $\mathbf{v} \in \mathbb{R}^{N_{\mathcal{M}} \times D_{\mathcal{M}}}$ given target protein as $\mathbf{p}$.

SBDD aims to discover a molecule $\mathbf{m}$ given protein and its binding site $\mathbf{p}$. During training, a set of protein–reference molecule pairs is provided. As a primary performance metric, external software such as AutoDock Vina (Eberhardt et al., 2021) is employed to estimate the binding affinity between the protein and generated molecules. Along with affinity scores, additional molecular properties (e.g., validity, synthesizability, and drug-likeness) can be evaluated to assess the quality of the generated compounds further.

## 3.2. Bayesian Flow Networks

Bayesian flow network (BFN) (Graves et al., 2023) is a class of generative models that integrate core ideas from both diffusion models (Ho et al., 2020) and Bayesian inference. Unlike traditional diffusion models that operate directly in the data space, BFNs perform their forward and reverse "diffusion" in the parameter space of the data distribution. This process ensures the generative process is continuous and differentiable regardless of whether data is discrete or continuous, making BFNs particularly suitable for molecule generation, where atom coordinates are continuous, and atom types are discrete. MolCRAFT (Qu et al., 2024) adopts the BFN as a backbone for conditional molecule generative model given proteins. Below, we outline the details of BFNs in the context of MolCRAFT.

At a high level, BFNs can be conceptualized as an iterative message exchange between two components: a sender and a receiver. In each round, the sender transmits a noisy sample to the receiver, akin to the forward diffusion process in a standard diffusion model. The receiver then attempts to infer the sender's parameters from the noisy input, effectively performing a reverse "denoising" step analogous to the reverse diffusion phase. This iterative cycle continues until reaching the final estimate of the data distribution in parameter space.

**Sender distribution** Formally, given a molecule $\mathbf{m} = [\mathbf{x}, \mathbf{v}]$, the sender injects continuous noise for both coordinates and types independently according to a predefined noise schedule. We denote the resulting noise-injected molecule by $\tilde{\mathbf{m}} = [\tilde{\mathbf{x}}, \tilde{\mathbf{v}}]$. The *sender distribution* is then factorized as:

$$p_S(\tilde{\mathbf{m}}|\mathbf{m}, t) = \prod_i^{N_\mathcal{M}} p(\tilde{\mathbf{x}}^{(i)}|\mathbf{x}^{(i)}, \alpha_t^{\mathbf{x}}) \prod_i^{N_\mathcal{M}} p(\tilde{\mathbf{v}}^{(i)}|\mathbf{v}^{(i)}, \alpha_t^{\mathbf{v}}), \quad (1)$$

where $\alpha_t$ is an accuracy parameter such that the sender samples are entirely uninformative about $\mathbf{x}$ when $\alpha = 0$ and become more informative as $\alpha$ increases. One can use different schedules for the coordinate $\alpha_t^{\mathbf{x}}$ and atom type $\alpha_t^{\mathbf{v}}$.

**Receiver distribution** At time t, the receiver uses parameters $\boldsymbol{\theta}_{t-1}$ from the previous step and the noisy sample $\tilde{\mathbf{m}}$ from the sender to reconstruct a denoised $\mathbf{m}$. Let $\boldsymbol{\Phi}(\boldsymbol{\theta}_{t-1}, \mathbf{p}, t)$ be an SE(3)-equivariant neural network that outputs $\hat{\mathbf{m}}$ from parameters $\boldsymbol{\theta}_{t-1}$, the protein pocket $\mathbf{p}$, and time $t$. Given that the distribution and noise schedule of the sender distribution is known, the receiver distribution is defined as

$$p_R(\tilde{\mathbf{m}} \mid \boldsymbol{\theta}_{t-1}, \mathbf{p}, t) = p_S(\tilde{\mathbf{m}} \mid \boldsymbol{\Phi}(\boldsymbol{\theta}_{t-1}, \mathbf{p}, t), \alpha_t). \quad (2)$$

Hence the receiver distribution becomes close to the sender distribution as $\boldsymbol{\Phi}$ accurately reconstructs the original molecule $\mathbf{m}$.

**Bayesian update** To ensure that distribution parameters $\boldsymbol{\theta}_t$ can be updated in a tractable way, BFNs employ Bayesian inference. Given the previous parameter $\boldsymbol{\theta}_{t-1}$ and the noisy sample $\tilde{\mathbf{m}}_t$, the updated parameter $\boldsymbol{\theta}_t$ can be obtained via a *Bayesian update function* $h(\boldsymbol{\theta}_{t-1}, \tilde{\mathbf{m}}_t, \alpha_t) = \boldsymbol{\theta}_t$. The update function infers the posterior after observing noisy sample $\tilde{\mathbf{m}}_t$ from the sender distribution $p_S(\tilde{\mathbf{m}}_t|\mathbf{m}, \alpha_t)$ and an *input distribution* $p_I(\mathbf{m}|\boldsymbol{\theta}_{t-1})$, often selected as a conjugate prior of the sender distribution.

This induces the *Bayesian update distribution* $p_U$, which is a push-forward distribution of $p_S$ under transformation $h$, i.e., $p_U(\boldsymbol{\theta}') := p_S(\{\tilde{\mathbf{m}} \mid h(\tilde{\mathbf{m}}) = \boldsymbol{\theta}'\})$, where we omit the conditions for brevity. By marginalizing out $\boldsymbol{\theta}_{1:t-1}$, one obtains the *Bayesian flow distribution*:

$$p_F(\boldsymbol{\theta}_t \mid \mathbf{m}, \mathbf{p}, t) = p_U(\boldsymbol{\theta}_t \mid \boldsymbol{\theta}_0, \mathbf{m}, \mathbf{p}, \alpha_{0:t}), \quad (3)$$

where $\boldsymbol{\theta}_0$ is the initial parameter used to define a simple distribution such as the standard normal for continuous or uniform for discrete. A key advantage of BFNs is that, with suitable noise injections and the input distribution, $\boldsymbol{\theta}_t$ can be calculated recursively without explicit simulation.

**Training objective** Training proceeds by minimizing the Kullback–Leibler divergence between the sender distribution $p_S$ and the receiver distribution $p_R$:

$$L(\mathbf{m}, \mathbf{p}) = \mathbb{E}_{t \sim U(1,T), \tilde{\mathbf{m}}_t \sim p_S, \boldsymbol{\theta}_{t-1} \sim p_F}[D_{\mathrm{KL}}(p_S \parallel p_R)]. \quad (4)$$

For the continuous-time extension of BFNs and additional technical details, we refer readers to Graves et al. (2023).

## 3.3. Direct Preference Optimization

Reinforcement learning with human feedback (RLHF) has been proposed to align the large language models with human preference (Ouyang et al., 2022). Let $\mathcal{D} = \{(\mathbf{c}, \mathbf{x}^w, \mathbf{x}^l)\}$ be a preference dataset, where $\mathbf{x}^w$ and $\mathbf{x}^l$ are winning and losing responses given prompt $\mathbf{c}$. In standard

RLHF pipelines, one first trains a reward model to capture these preferences and then uses policy optimization methods (e.g., PPO) to fine-tune the model such that it produces higher-scoring outputs.

Direct Preference Optimization (DPO) offers an alternative that bypasses the need for an explicit reward-model approximation in the training loop (Rafailov et al., 2024). Let $p_\theta(\mathbf{x})$ be a model to be fine-tuned. Given a reference model $p_{\text{ref}}$, DPO is derived from the optimal solution to the RLHF objective and can be expressed as

$$L_{\text{DPO}}(\theta) =$$
$$-\mathbb{E}_{\mathbf{c}, \boldsymbol{x}^w, \boldsymbol{x}^l} \left[ \log \sigma \left( \beta \log \frac{p_\theta(\boldsymbol{x}^w)}{p_{\text{ref}}(\boldsymbol{x}^w)} - \beta \log \frac{p_\theta(\boldsymbol{x}^l)}{p_{\text{ref}}(\boldsymbol{x}^l)} \right) \right], \tag{5}$$

where we omit the condition $\mathbf{c}$ for brevity. Minimizing this loss encourages the fine-tuned model $p_\theta$ to assign higher relative likelihood to the winning samples than to the losing samples, while remaining close to $p_{\text{ref}}$. Recently, Diffusion-DPO extends DPO to diffusion-based generative models by expanding the reward model over the entire diffusion trajectory, enabling preference-driven fine-tuning in multi-step generative processes (Wallace et al., 2023).

## 4. Multi-Reward Optimization for BFNs

In this section, we present our Direct Preference Optimization for Bayesian Flow Networks (BFN-DPO), an end-to-end framework for controllable molecule generation. Our method draws on recent advances in DPO and adapts these ideas to BFNs. We then extend DPO-BFN to handle *multiple* reward signals via a softmax-based normalization strategy and an uncertainty-regularized ensemble, allowing fair comparisons among rewards with different numerical scales.

### 4.1. Direct Preference Optimization on BFNs

We consider a preference dataset $\mathcal{D} = \{(\mathbf{p}, \mathbf{m}^w, \mathbf{m}^l)\}$, where each triplet consists of a protein target $\mathbf{p}$, a "winning" molecule $\mathbf{m}^w$, and a "losing" molecule $\mathbf{m}^l$. These preferences are determined by pre-computed reward values (e.g., binding affinity). We denote by $p_{\text{ref}}(\cdot)$ and $p_\phi(\cdot)$ the reference and fine-tuned BFN models, respectively. Drawing on Diffusion-DPO (Wallace et al., 2023), we define the BFN-DPO objective for a single preference pair $(\mathbf{m}^w, \mathbf{m}^l)$ as

$$\mathcal{L}_{\text{BFN-DPO}}(\phi) = -\mathbb{E} \Big[ \log \sigma \Big( \beta \log \frac{p_\phi(\hat{\mathbf{m}}_{1:T}^w \mid \boldsymbol{\theta}_{0:T-1}^w)}{p_{\text{ref}}(\hat{\mathbf{m}}_{1:T}^w \mid \boldsymbol{\theta}_{0:T-1}^w)}$$
$$- \beta \log \frac{p_\phi(\hat{\mathbf{m}}_{1:T}^l \mid \boldsymbol{\theta}_{0:T-1}^l)}{p_{\text{ref}}(\hat{\mathbf{m}}_{1:T}^l \mid \boldsymbol{\theta}_{0:T-1}^l)} \Big) \Big], \tag{6}$$

where $\hat{\mathbf{m}}_t$ is a noisy sample drawn at time step $t$ (the "receiver" output), governed by the BFN's sender/receiver distributions, and $\boldsymbol{\theta}_t$ are parameters sampled from the Bayesian flow distribution $p_F$, which performs Bayesian updates in parameter space.

Unlike Diffusion-DPO, the reconstruction at time $t$ depends on the parameter $\boldsymbol{\theta}_{t-1}$. Expanding the ratio into a sum over time steps with Jensen's inequality, one obtains an upper bound involving differences of KL terms at each time step. By approximating the receiver distribution in expectation with the sender distribution, we can obtain the following loss:

$$\mathcal{L}_{\text{BFN-DPO}}(\phi) = -\mathbb{E}\Big[\mathcal{L}_t^{\mathbf{x}} + \mathcal{L}_t^{\mathbf{v}}\Big], \tag{7}$$

which is separated into two parts: atom loss $\mathcal{L}_t^{\mathbf{x}}$ and type loss $\mathcal{L}_t^{\mathbf{v}}$. A complete derivation is provided in Appendix A.

Let $\mathbf{x}^w$ and $\mathbf{x}^l$ be the clean coordinates for the winning and losing molecules at time $t = 0$. Then the atom loss has the form of

$$\mathcal{L}_t^{\mathbf{x}} = -\mathbb{E}\Big[\log \sigma\Big(-\beta T \frac{\alpha_t^{\mathbf{x}}}{2} \big[\Delta_\phi^w - \Delta_\phi^l\big]\Big)\Big], \tag{8}$$

with

$$\Delta_\phi^w = \big\| \mathbf{x}^w - \Phi_\phi^{\mathbf{x}}(\boldsymbol{\theta}_{t-1}^{\mathbf{x},w}) \big\|^2 - \big\| \mathbf{x}^w - \Phi_{\text{ref}}^{\mathbf{x}}(\boldsymbol{\theta}_{t-1}^{\mathbf{x},w}) \big\|^2,$$

and analogously for $\Delta_\phi^l$. Here, $\Phi_\phi^{\mathbf{x}}(\cdot)$ and $\Phi_{\text{ref}}^{\mathbf{x}}(\cdot)$ generate coordinate predictions under the fine-tuned and reference BFNs, and $\alpha_t^{\mathbf{x}}$ is a concentration parameter for the coordinates at step $t$.

For the discrete atom types, we have a similar construction based on KL divergences. Each "winning" and "losing" noisy type $\hat{\mathbf{v}}_t^w$ and $\hat{\mathbf{v}}_t^l$ is compared to the true one-hot type vector $\mathbf{v}^w$ or $\mathbf{v}^l$. The derivation yields:

$$\mathcal{L}_t^{\mathbf{v}} = -\mathbb{E}\Big[\log \sigma\Big(-\beta T \big[(\Delta_\phi^w)_{\mathbf{v}} - (\Delta_\phi^l)_{\mathbf{v}}\big]\Big)\Big], \tag{9}$$

where

$$(\Delta_\phi^w)_{\mathbf{v}} =$$
$$\ln p_S(\hat{\mathbf{v}}_t^w | \mathbf{v}^w, \alpha_t^{\mathbf{v}}) - \ln p_R(\hat{\mathbf{v}}_t^w | \Phi_\phi^{\mathbf{v}}(\boldsymbol{\theta}_{t-1}^{\mathbf{v},w}), \alpha_t^{\mathbf{v}})$$
$$- [\ln p_S(\hat{\mathbf{v}}_t^w | \mathbf{v}^w, \alpha_t^{\mathbf{v}}) - \ln p_R(\hat{\mathbf{v}}_t^w | \Phi_{\text{ref}}^{\mathbf{v}}(\boldsymbol{\theta}_{t-1}^{\mathbf{v},w}), \alpha_t^{\mathbf{v}})],$$

and similarly for $(\Delta_\phi^l)_{\mathbf{v}}$. In practice, these log-likelihood terms could be implemented as cross-entropy or KL divergences for categorical predictions.

By minimizing $\mathcal{L}_{\text{BFN-DPO}}(\phi)$, the model $p_\phi$ is driven to generate noisy samples $\hat{\mathbf{m}}_t^w$ that better align with automated preferences than $\hat{\mathbf{m}}_t^l$, all while remaining close to $p_{\text{ref}}$.

## 4.2. Multi-Reward Normalization

While one can fine-tune a Bayesian Flow Network (BFN) solely using a single reward signal (e.g., validity), doing so may degrade other desirable properties like binding affinity. To address this limitation, we adopt a multi-reward BFN-DPO scheme inspired by Kim et al. (2024), where multiple reward signals (e.g., binding affinity, validity) are simultaneously optimized.

Suppose we have $K$ reward functions that evaluate a molecule–protein pair $(\mathbf{p}, \mathbf{m})$. Since these rewards may have different numerical ranges and may be preferred in different directions (e.g., negative for binding affinity, positive for strain energy), direct averaging can be misleading. To handle scale discrepancies, we normalize each reward via a softmax over a mini-batch of size $B$:

$$\hat{r}_i^{(j)} = \frac{\exp\big(f_j(r_i^{(j)})\big)}{\sum_{b=1}^{B} \exp\big(f_j(r_b^{(j)})\big)}, \qquad (10)$$

where $r_i^{(j)}$ is the $j$-th reward for the $i$-th molecule, and $f_j(\cdot)$ is a simple transformation based on the $j$-th reward, so that higher values consistently indicate better rewards across different metrics. This maps each reward to the $(0, 1)$ interval, ensuring that distinct reward signals become comparable despite inherently different numerical ranges.

After normalization, one still must combine the m rewards into a single score. Following Kim et al. (2024), we reduce the influence of high-variance reward profiles by subtracting a variance-based penalty. Specifically, let $\mu_{\hat{r}_i}$ be the mean of the normalized rewards for molecule $i$. The final multi-reward score for that molecule is

$$\bar{r}_i = \mu_{\hat{r}_i} - \gamma \frac{1}{K} \sum_{j=1}^{K} (\hat{r}_i^{(j)} - \mu_{\hat{r}_i})^2 \qquad (11)$$

where $\gamma$ is a hyperparameter controlling how strongly to penalize high reward variance. This strategy promotes molecules that achieve balanced performance across all reward dimensions, rather than excelling at some while failing at others. More details on the hyperparameters can be found in Appendix B.

We further stabilize training by adopting the E²PO, a variant of DPO (Gu et al., 2024), to avoid excessive focus on winning samples when their advantage is already large. Concretely, we replace the standard BFN-DPO loss $\mathcal{L}_{\text{BFN-DPO}}$ with

$$\begin{aligned} \mathcal{L}_{\text{BFN-E}^2\text{PO}} = - \; \mathbb{E}\Big[ & \sigma(\bar{r}_w - \bar{r}_l) \, \mathcal{L}_{\text{BFN-DPO}} \\ & + (1 - \sigma(\bar{r}_w - \bar{r}_l)) (2 - \mathcal{L}_{\text{BFN-DPO}}) \Big], \end{aligned} \qquad (12)$$

where $\bar{r}_w$ and $\bar{r}_l$ denote the final multi-reward scores for the winning and losing molecules, respectively. If the difference $\bar{r}_w - \bar{r}_l$ becomes too large (close to 1), the second term is nullified; thus, over-optimizing already "high-scoring" winners is discouraged. In effect, E²PO enables more balanced training and helps prevent mode collapse onto a narrow set of winning samples.

## 5. Experiment

In this section, we provide the result of the multi-object optimization approach with empirical experiments. Due to the space limit, missing implementation details are provided in Appendix B.

### 5.1. Experimental Setting

**Datasets** We fine-tune a pretrained model on the Cross-Docked dataset (Francoeur et al., 2020). The CrossDocked dataset comprises 100K training pairs and 100 target proteins for testing. For each protein, win-lose pairs are selected based on multi-reward scores described in Section 4.2 from molecules generated by the pretrained model. These pairs serve as input for fine-tuning via our proposed BFN-DPO procedure. We generate 100 new molecules per target protein in the test set for evaluation.

**Baselines** We compare our method to the following baselines: AR (Luo et al., 2021) and Poket2Mol (Peng et al., 2022) generate molecules autoregressive, and FLAG (Zhang & Liu, 2023) generates at the fragment level. TargetDiff (Guan et al., 2023), DecompDiff (Guan et al., 2024), DecompOpt (Zhou et al., 2024), and IPDiff (Huang et al., 2024) are diffusion-based generative models. IPDiff incorporates protein-ligand interactions as a priori in the forward process. MolCRAFT (Qu et al., 2024) employs BFNs, where noise and denoise processes are performed in a fully continuous parameter space. AliDiff (Gu et al., 2024) applies a single reward optimization to improve the binding affinity based on IPDiff.

**Evaluation metrics** We evaluate the generated molecules from three different perspectives: binding affinity, conformation validity, and molecular properties. For binding affinity, we use Vina Score, Vina Min, and Vina Dock measured by AutoDock Vina (Eberhardt et al., 2021) following the setup used in Ragoza et al. (2022). The Vina Score measures the binding affinity of the generated molecule as is. Vina Min measures the affinity after optimizing the molecular structure without changing the docking position. Vina Dock measures the binding affinity after re-docking the molecule; hence, reconsider the molecule's position and orientation. Low scores indicate stronger protein-ligand binding affinity. We report average and median scores for all Vina metrics. For conformation validity, we report strain energy (SE) measuring molecule conformation energy and protein-ligand

*Table 1.* Performance comparison across different reward methods. We mark the best, and the second-best performances in **bold** and underline, respectively. (↓)/(↑) indicate whether a smaller/larger number is better, respectively.

| Method | Binding Affinity | | | | | | Conformation Stability | | | | Molecular Properties | | Avg. size |
| --- | --- | --- | --- | --- | --- | --- | --- | --- | --- | --- | --- | --- | --- |
| | Vina Score (↓) | | Vina Min (↓) | | Vina Dock (↓) | | SE (↓) | | | Clash (↓) | SA (↑) | QED (↑) | |
| | Avg. | Med. | Avg. | Med. | Avg. | Med. | 25% | 50% | 75% | Avg. | Avg. | Avg. | |
| Reference | -6.36 | -6.46 | -6.71 | -6.49 | -7.45 | -7.26 | 1.98 | 3.98 | 6.23 | 5.51 | 0.73 | 0.48 | 22.8 |
| AR | -5.75 | -5.64 | -6.18 | -5.88 | -6.75 | -6.62 | 2.79 | 7.39 | 16.34 | **4.46** | 0.64 | 0.51 | 17.7 |
| Pocket2Mol | -5.14 | -4.70 | -6.42 | -5.82 | -7.15 | -6.79 | **0.22** | **1.50** | **4.82** | 6.24 | **0.74** | **0.57** | 17.7 |
| FLAG | 16.48 | 4.53 | 1.21 | -4.04 | -5.63 | -6.61 | 1.32 | 4.96 | 16.84 | 40.83 | 0.70 | 0.49 | 21.5 |
| TargetDiff | -5.47 | -6.30 | -6.64 | -6.83 | -7.80 | -7.91 | 7.48 | 16.85 | 33.70 | 10.84 | 0.58 | 0.48 | 24.2 |
| DecompDiff | -5.19 | -5.27 | -6.03 | -6.00 | -7.03 | -7.16 | 4.17 | 10.81 | 22.77 | 8.16 | 0.66 | 0.51 | 21.2 |
| DecompOpt | -5.67 | -6.04 | -7.04 | -7.09 | -8.39 | -8.43 | 12.34 | 21.37 | 37.69 | 14.56 | 0.61 | 0.45 | 29.4 |
| IPDiff | -6.70 | -7.53 | -7.63 | -7.71 | -8.41 | -8.39 | 10.30 | 27.05 | 65.49 | 8.79 | 0.61 | 0.52 | 24.3 |
| AliDiff | -6.97 | **-7.75** | -7.87 | **-7.94** | **-8.65** | **-8.64** | 10.09 | 27.60 | 67.00 | 8.76 | 0.56 | 0.50 | 24.4 |
| MolCRAFT | -6.59 | -7.04 | -7.27 | -7.26 | -7.92 | -8.01 | 3.10 | 7.92 | 14.87 | 7.09 | 0.69 | 0.50 | 22.7 |
| Ours | **-7.18** | -7.38 | **-7.89** | -7.77 | -8.62 | **-8.64** | 2.05 | 5.56 | 11.62 | 6.69 | **0.74** | 0.55 | 22.8 |

clash measuring the number of potential overlaps between ligand atoms and protein (Harris et al., 2023). For molecular properties, we use synthetic accessibility (SA) (Ertl & Schuffenhauer, 2009) and quantitative estimation of drug-likeness (QED) (Bickerton et al., 2012).

**Multi-reward optimization** We choose MolCRAFT (Qu et al., 2024) as a backbone for the multi-reward optimization. In the following analysis, we report the results of the fine-tuned model with Vina Score, SE, and QED. Although it is possible to optimize all seven evaluation metrics together with BFN-DPO, we find that this generally degrades the model performance. A more detailed analysis of reward selection is provided in Section 5.4.

### 5.2. Results

Table 1 shows the overall performance of ligand molecules generated by our method and baselines. Our approach achieves four best and four second-best results in eight metrics, indicating robust and well-rounded performance in SBDD. While autoregressive models ensure stable conformations and molecular properties but struggle with binding affinity, and diffusion-based models achieve high binding affinity at the cost of conformation stability and synthetic accessibility, MolCRAFT with BFNs balances both effectively. Qu et al. (2024) observes that binding affinity scores can be artificially boosted by generating larger molecules, which present more surface area for binding. The average size of our generated molecules remains comparable to those produced by MolCRAFT, indicating that no affinity hacking occurs. Note that smaller molecules often excel in SE and SA, which helps explain why AR and Pocket2Mol, both of which generate relatively smaller ligands, report particularly high SE and SA scores.

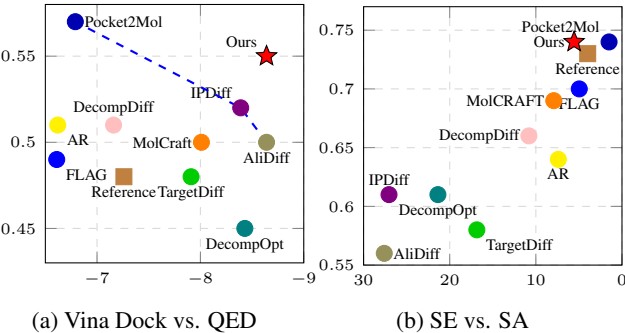

(a) Vina Dock vs. QED      (b) SE vs. SA

*Figure 2.* Comparison between the median Vina docking score and QED, and SE and SA.

Figure 1 illustrates how different models balance their average Vina Dock scores against SA and SE. In both subplots, previous approaches occupy a curve where improving one metric typically comes at the expense of the other. In contrast, our multi-reward method moves beyond this frontier by achieving stronger binding affinity while simultaneously preserving higher SA and lower SE. This outcome highlights the effectiveness of our framework in expanding the Pareto front for multi-objective structure-based drug design.

We further illustrate the comparison between Vina Dock and QED in Figure 2a and SE and SA in Figure 2b. A similar trend can be observed from the Vina Dock and QED, but no clear trade-off between SE and SA is observable. Instead, we observe a strong correlation between SE and SA.

### 5.3. Analysis

**Validity of generated molecules** Although the conformation stability and molecular property metrics show the validness of generated molecules by our method, we con-

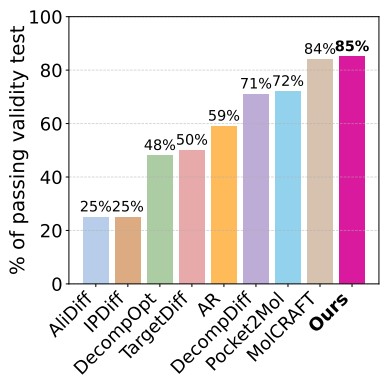

(a) The percentage of valid molecules.

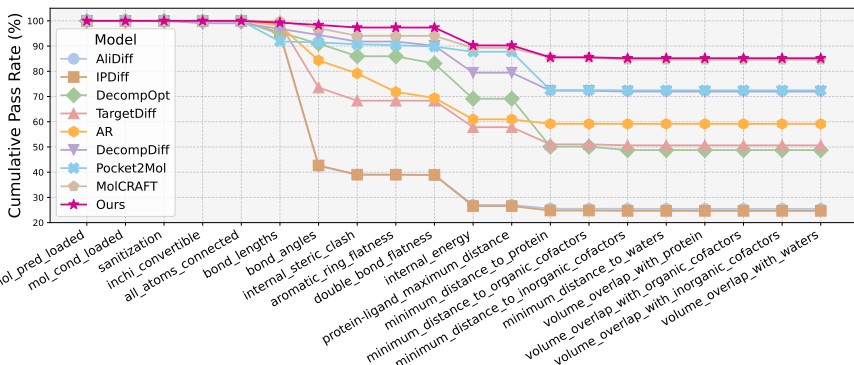

(b) Cumulative chart showing the remaining percent of valid molecules after each validity test.

*Figure 3.* Validity check of the generated molecules from different models through PoseBuster (Buttenschoen et al., 2024).

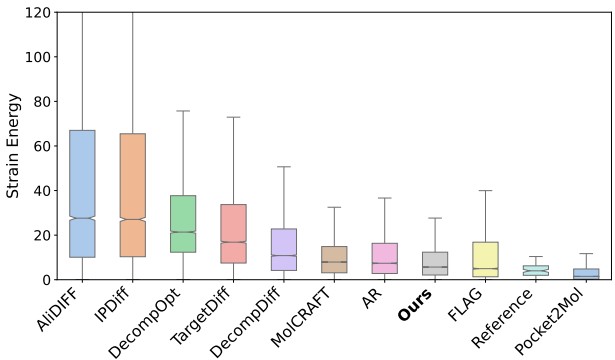

*Figure 4.* Comparison of strain energy (SE) distributions for molecules generated by different methods.

*Table 2.* Comparison between different opitmization approaches. GS is short for gradient surgery. A model fine-tuned with multi-rewards shows better binding affinity scores than the model with gradient surgery.

| | Binding Affinity | Conformation Stability | | Molecular Properties | |
|---|---|---|---|---|---|
| | Vina Score Med. (↓) | SE Med. (↓) | Clash Avg. (↓) | SA Avg. (↑) | QED Avg. (↑) |
| Reference | -7.46 | 3.96 | 5.51 | 0.73 | 0.48 |
| Pretrained | -7.04 | 7.62 | 7.09 | 0.69 | 0.50 |
| 3 GS | -6.80 | 9.69 | 7.18 | 0.66 | 0.50 |
| 6 GS | -6.87 | 8.62 | 7.11 | 0.69 | 0.52 |
| Multi-Stage | -7.16 | **5.15** | 7.54 | **0.74** | 0.54 |
| Ours | **-7.38** | 5.56 | **6.69** | **0.74** | **0.55** |

duct a more comprehensive validity check through the 20 validity tests proposed by PoseBuster (Buttenschoen et al., 2024). The test considers both intra-molecular and inter-molecular validity with the protein. Figure 3a shows the percentage of molecules that pass all validity tests on the generated 10K molecules for the test proteins. Our multi-reward method generates significantly more valid ligands compared to other generative models. AliDiff, a fine-tuned IPDiff (Huang et al., 2024) with Vina Dock as a single reward, is known to achieve the best binding performance so far. However, the model fails to generate valid molecules in many cases. Figure 3b shows which test each model fails to satisfy the validity over 20 validity tests. Notably, many models, including IPDiff and AliDiff, fail to satisfy proper bond angles between atoms of generated molecules. Some models fail to keep a proper distance from the protein.

**Strain Energy (SE)** Recent studies (Buttenschoen et al., 2024; Harris et al., 2023) caution that deep learning-based SBDD models may artificially inflate binding scores by producing ligands with unrealistically high SE. To examine

this issue, Figure 4 presents the SE distributions for different methods. Lower SE values correspond to more physically plausible, stable 3D conformations. Our method clearly generates molecules with lower SE distributions, indicating structurally sound ligands. These improvements do not come at the expense of binding affinity.

**Synthetic Accessibility (SA)** Figure 5 presents the median SA scores of molecules generated by AliDiff, IPDiff, and MolCRAFT with reference molecules across target proteins in the test set. The compared models are recent state-of-the-art models for binding affinity. The results show that our model consistently generates molecules with high SA across all targets. We also report the proportion of proteins for which each model generated molecules with the highest median SA score. Our model obtains the best SA scores for 44% of proteins.

### 5.4. Ablation Studies

**Different optimization approaches** Multi-reward optimization can be framed as a multi-task learning problem

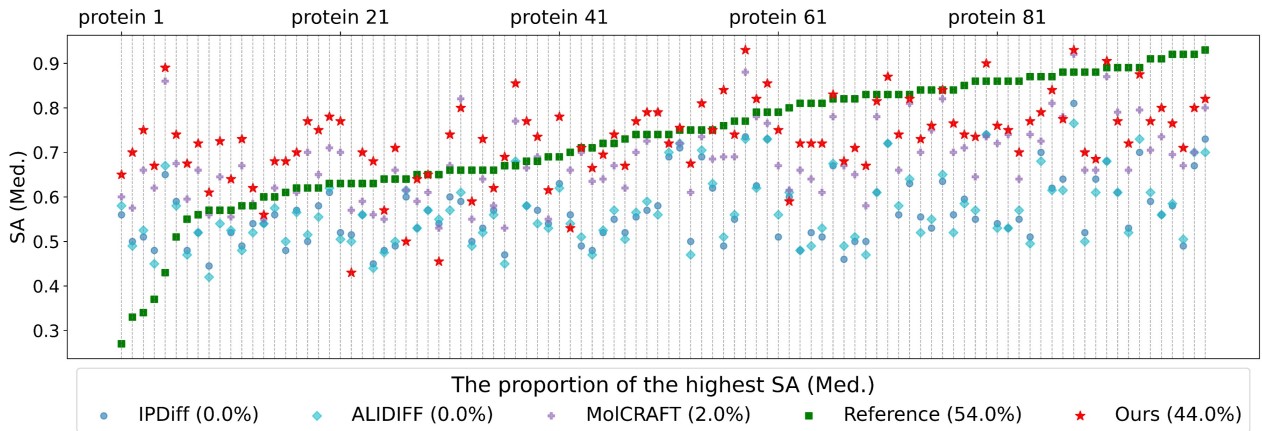

*Figure 5.* The median Synthetic Accessibility (SA) scores of the molecules generated for each of the 100 target proteins in the test set. We compare our model to the top three models from the highest median binding affinity (AliDiff, IPDiff, MolCRAFT), as well as the reference. The values in parentheses represent the proportion of the best SA score for each protein

*Table 3.* Comparison based on the number of rewards. A model fine-tuned with three rewards shows better binding affinity scores than the model with all seven rewards.

| # Rewards | | Binding Affinity | | | Conformation Stability | | Drug-like Properties | |
|---|---|---|---|---|---|---|---|---|
| | | Vina Score Avg. ($\downarrow$) | Vina Min Avg. ($\downarrow$) | Vina Dock Avg. ($\downarrow$) | SE Med. ($\downarrow$) | Clash Avg. ($\downarrow$) | SA Avg. ($\uparrow$) | QED Avg. ($\uparrow$) |
| No | Pretrained | -6.59 | -7.27 | -7.92 | 7.62 | 7.09 | 0.69 | 0.50 |
| Single | Score | -6.63 | -7.36 | -8.01 | 6.85 | 7.25 | 0.71 | 0.53 |
| | Min | -6.61 | -7.43 | -8.02 | 6.88 | 7.36 | 0.71 | 0.53 |
| | Dock | -6.63 | -7.40 | -8.1 | 6.99 | 7.34 | 0.71 | 0.53 |
| | SE | -6.51 | -7.25 | -7.79 | 6.79 | 7.40 | 0.70 | 0.50 |
| | Clash | -6.44 | -7.09 | -7.58 | 9.87 | 7.16 | 0.66 | 0.48 |
| | QED | -6.45 | -7.31 | -7.80 | 6.13 | 7.38 | 0.72 | 0.54 |
| | SA | -6.50 | -7.29 | -7.80 | 7.33 | 7.47 | 0.71 | 0.53 |
| Multi (Ours) | Dock+SE+QED | **-7.18** | **-7.89** | **-8.62** | **5.56** | **6.69** | **0.74** | **0.55** |
| All | 7 rewards | -6.50 | -7.28 | -7.86 | 7.22 | 7.28 | 0.53 | 0.70 |

when each reward is treated as a separate task. To explore this perspective, we adopt the widely used multi-task optimization technique known as *gradient surgery* (Yu et al., 2020). Specifically, we apply the gradient surgery to DPO losses of Vina score, SE, and QED (3GS). Since the DPO loss consists of atom and type losses as in Equation (7), we further apply the gradient surgery to these losses separately, resulting in six losses in total (6GS). In addition, we examine a *multi-stage* optimization procedure, wherein the model is iteratively fine-tuned using newly generated molecules from each preceding iteration. We report the result of the two-stage approach.

Table 2 compares the performance of each approach to our uncertainty-based reward optimization. We only report the Vina Score among all Vina metrics, but they share similar trends. Neither the gradient surgery nor the multi-stage

approach outperforms our uncertainty-based reward optimization. The gradient surgery methods perform worse than the reference model in all metrics except QED.

**Multi-reward selection**    Table 3 show the performances with different choices of metrics as rewards. Trying to optimize *all* seven rewards at once underperforms in several categories, implying the complex loss landscape with all rewards. In general, optimizing Vina metrics shows a strong correlation, i.e., optimizing one Vina metric can improve the performance of the other two Vina metrics. The optimization over QED improves the performance of SA, leading to our final choice of rewards. Note that Clash cannot be improved even by the corresponding single reward. We leave the improvement of Clash for future work.

# 6. Conclusion

In this work, we propose a multi-reward optimization framework to enhance deep learning-based structure-based drug design. Our method addresses the limitations of previous approaches by fine-tuning generative models to optimize multiple attributes, including binding affinity, validity, and drug-likeness. By utilizing direct preference optimization within a Bayesian flow network and integrating a reward normalization scheme, we enable the generation of more realistic ligands with improved binding affinity. Experimental results demonstrate that our approach outperforms the baseline models, offering a more robust solution for drug discovery tasks.

# Acknowledgements

This work was partly supported by Institute of Information & communications Technology Planning & Evaluation (IITP) grant funded by the Korea government (MSIT) (RS-2019-II191906: Artificial Intelligence Graduate School Program (POSTECH), RS-2024-00457882: Artificial Intelligence Research Hub Project) and Korea Evaluation Institute of Industrial Technology (KEIT) grant funded by the Korea government (MOTIE, Korea) and the National Research Foundation of Korea (NRF) grant funded by the Korea government (MSIT) (RS-2024-00337955, RS-2023-00217286)

# Impact Statement

This paper presents work whose goal is to advance the field of Machine Learning. There are many potential societal consequences of our work, none which we feel must be specifically highlighted here.

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

## A. Complete Derivation of BFN-DPO

Given a preference molecular dataset as $\mathcal{D} = \{(\mathbf{p}, \mathbf{m}^w, \mathbf{m}^l)\}$ where $\mathbf{p}$ is the target proteins and $\mathbf{m}^w, \mathbf{m}^l$ are winning-losing pair of molecules based on pre-computed rewards. With the similar argument used in Diffusion-DPO, one can define an objective for BFN over the entire timesteps as:

$$L_{\text{BFN-DPO}}(\phi) = -\mathbb{E}_{\mathbf{m}^w,\mathbf{m}^l,\mathbf{p}\sim\mathcal{D}} \log \sigma \left( \beta\mathbb{E}_{p_\phi(\mathbf{m}^w_{1:T}|\mathbf{m}^w)p_\phi(\mathbf{m}^l_{1:T}|\mathbf{m}^l)} \left[ \log \frac{p_\phi(\mathbf{m}^w_{1:T})}{p_{\text{ref}}(\mathbf{m}^w_{1:T})} - \log \frac{p_\phi(\mathbf{m}^l_{1:T})}{p_{\text{ref}}(\mathbf{m}^l_{1:T})} \right] \right) \tag{13}$$

$$= -\mathbb{E}_{\mathbf{m}^w,\mathbf{m}^l,\mathbf{p}\sim\mathcal{D}} \log \sigma \left( \beta\mathbb{E}_{\substack{p_\phi(\mathbf{m}^w_{1:T}|\mathbf{m}^w,\theta^w_{0:T-1})p_\phi(\theta^w_{0:T-1}|\mathbf{m}^w), \\ p_\phi(\mathbf{m}^l_{1:T}|\mathbf{m}^l,\theta^l_{0:T-1})p_\phi(\theta^l_{0:T-1}|\mathbf{m}^l)}} \left[ \log \frac{p_\phi(\mathbf{m}^w_{1:T})}{p_{\text{ref}}(\mathbf{m}^w_{1:T})} - \log \frac{p_\phi(\mathbf{m}^l_{1:T})}{p_{\text{ref}}(\mathbf{m}^l_{1:T})} \right] \right) \tag{14}$$

$$= -\mathbb{E}_{\mathbf{m}^w,\mathbf{m}^l,\mathbf{p}\sim\mathcal{D}} \log \sigma \left( \beta\mathbb{E}_{\substack{p_\phi(\mathbf{m}^w_{1:T}|\mathbf{m}^w,\theta^w_{0:T-1}) \\ p_\phi(\theta^w_{0:T-1}|\mathbf{m}^w) \\ p_\phi(\mathbf{m}^l_{1:T}|\mathbf{m}^l,\theta^l_{0:T-1}) \\ p_\phi(\theta^l_{0:T-1}|\mathbf{m}^l)}} \left[ \log \frac{p_\phi(\mathbf{m}^w_{1:T}|\theta^w_{0:T-1})p_\phi(\theta^w_{0:T-1}|\mathbf{m}^w)}{p_{\text{ref}}(\mathbf{m}^w_{1:T}|\theta^w_{0:T-1})p_\phi(\theta^w_{1:T-1}|\mathbf{m}^w)} \right. \right.$$

$$\left. \left. - \log \frac{p_\phi\left(\mathbf{m}^l_{1:T}\right) p_\phi(\theta^l_{1:T-1}|\mathbf{m}^l)}{p_{\text{ref}}(\mathbf{m}^l_{1:T})p_\phi(\theta^l_{0:T-1}|\mathbf{m}^l)} \right] \right) \tag{15}$$

$$= -\mathbb{E}_{\mathbf{m}^w,\mathbf{m}^l,\mathbf{p}\sim\mathcal{D}} \log \sigma \left( \beta\mathbb{E}_{\substack{p_S(\mathbf{m}^w_t|\mathbf{m}^w,\alpha_t)p_F(\theta^w_{t-1}|\mathbf{m}^w) \\ p_S(\mathbf{m}^l_t|\mathbf{m}^l,\alpha_t)p_F(\theta^l_{t-1}|\mathbf{m}^l)}} \left[ \sum_{t=1}^T \left( \log \frac{p_\phi(\mathbf{m}^w_t|\theta^w_{t-1})}{p_{\text{ref}}(\mathbf{m}_t|\theta^w_{t-1})} - \log \frac{p_\phi(\mathbf{m}^l_t|\theta^l_{t-1})}{p_{\text{ref}}(\mathbf{m}_t|\theta^l_{t-1})} \right) \right] \right) + C, \tag{16}$$

where we approximate the intractable $p_\phi(\theta|\mathbf{m}^w)$ via Bayesian flow distribution $p_F$ and $p_\phi(\mathbf{m}^w_{1:T} \mid \mathbf{m}^w, \theta^w_{0:T-1})$ via the sender distribution. We omit the conditioning on the proteins $\mathbf{p}$ for brevity.

By applying Jensen's inequality and dropping constant, we have

$$L_{\text{BFN-DPO}}(\phi) = -\mathbb{E}_{\substack{\mathbf{m}^w,\mathbf{m}^l,\mathbf{p}\sim\mathcal{D},t\sim U(1,T), \\ p_F(\theta^w_i|\mathbf{m}^w),p_F(\theta^l_i|\mathbf{m}^l)}} \log \sigma \left( \beta T\mathbb{E}_{\substack{p_S(\mathbf{m}^w_i) \\ p_S(\mathbf{m}^l_i)}} \left[ \left( \log \frac{p_\phi(\mathbf{m}^w_i|\theta^w_{i-1})}{p_{\text{ref}}(\mathbf{m}^w_i|\theta^w_{i-1})} - \log \frac{p_\phi(\mathbf{m}^l_i|\theta^l_{i-1})}{p_{\text{ref}}(\mathbf{m}^l_i|\theta^l_{i-1})} \right) \right] \right) \tag{17}$$

Expanding the equation further gives us

$$\mathcal{L}_{\text{BFN-DPO}}(\phi) = -\mathbb{E}\left[ \log \sigma \left( -\beta T \left( \mathbb{D}_{\text{KL}}\left(p_S(\mathbf{m}^w_t \mid \mathbf{m}^w) \mid p_\phi\left(\mathbf{m}^w_t \mid \theta^w_{t-1}\right) - \mathbb{D}_{\text{KL}}\left(p_S(\mathbf{m}^w_t \mid \mathbf{m}^w) \mid p_{\text{ref}}(\mathbf{m}^w_t \mid \theta^w_{t-1})\right) \right. \right. \right. \right.$$

$$\left. \left. \left. \left. - \mathbb{D}_{\text{KL}}\left(p_S(\mathbf{m}^l_t \mid \mathbf{m}^l) \mid p_\phi\left(\mathbf{m}^l_t \mid \theta^l_{t-1}\right)\right) + \mathbb{D}_{\text{KL}}\left(p_S(\mathbf{m}^l_t \mid \mathbf{m}^l) \mid p_{\text{ref}}(\mathbf{m}^l_t \mid \theta^l_{t-1})\right) \right) \right) \right]. \tag{18}$$

The latent parameters $\theta_t = [\theta^x_t, \theta^v_t]$, from which the noisy samples $\mathbf{m}_t = [\mathbf{x}_t, \mathbf{v}_t]$ are drawn with the noise factors $\alpha_t = [\alpha^x_t, \alpha^v_t]$, are assumed to follow distinct continuous distributions for each atom coordinates $\mathbf{x}$ and atom types $\mathbf{v}$. With the neural network $\Phi$ representing the predicted atom coordinates and types as $[\Phi^x, \Phi^v]$, we can reformulate each of the above KL divergence terms as:

$$\mathbb{D}_{\text{KL}}\left(p_S(\mathbf{m}_t \mid \mathbf{x}) \mid p(\mathbf{m}_t \mid \theta_{t-1})\right)$$
$$= \mathbb{D}^x_{\text{KL}}\left(p_S(\mathbf{x}_t \mid \mathbf{x}) \mid p(\mathbf{x}_t \mid \theta^x_{t-1})\right) + \mathbb{D}^v_{\text{KL}}\left(p_S(\mathbf{v}_t \mid \mathbf{v}) \mid p(\mathbf{v}_{t-1} \mid \theta^v_{t-1})\right) \tag{19}$$
$$= \frac{\alpha^x_t}{2}\|\mathbf{x} - \Phi^x(\theta)\|^2 + \left( \log p_S(\mathbf{v}_t \mid \mathbf{v}, \alpha^v_t) - \log p_R(\mathbf{v}_t \mid \Phi^v(\theta_{t-1}), \alpha^v_t) \right). \tag{20}$$

As a result, we can show that the objective can be represented as a summation of the atom coordinates and atom type preference losses as follows:

$$\mathcal{L}_{\text{BFN-DPO}}(\phi) = -\mathbb{E}_{(\mathbf{p},\mathbf{m}^w,\mathbf{m}^l)\sim\mathcal{D},t\sim[0,T],\theta^w_t\sim p_F(\cdot|\mathbf{m}^w,\mathbf{p};t),\theta^l_t\sim p_F(\cdot|\mathbf{m}^l,\mathbf{p};t)} \left[ \mathcal{L}^x_{t-1} + \mathcal{L}^v_{t-1} \right]. \tag{21}$$

The atom coordinates loss is designed to directly train the model output $\Phi_0^{\mathbf{x}}(\theta_t^{\mathbf{x}})$ to approximate the clean coordinate values $\mathbf{x}_0$. The concentration parameter $\alpha$ for each time $t$ is multiplied during training:

$$
\mathcal{L}^{\mathbf{x}} = \underset{\substack{(\mathbf{p}, \mathbf{x}^w, \mathbf{x}^l) \sim \mathcal{D}, t \sim [0, T], \\ \theta_t^{\mathbf{x}, w} \sim p_F(\cdot | \mathbf{x}_t^w, \mathbf{p}; t), \theta_t^{\mathbf{x}, l} \sim p_F(\cdot | \mathbf{x}_t^l, \mathbf{p}; t)}}{-\mathbb{E}} \left[ \log \sigma \left( - \beta T \left( \frac{\alpha_t^{\mathbf{x}}}{2} \right) \left( \| \mathbf{x}^w - \Phi_\phi^{\mathbf{x}}(\theta_{t-1}^{\mathbf{x}, w}) \|^2 - \| \mathbf{x}^w - \Phi_{\text{ref}}^{\mathbf{x}}(\theta_{t-1}^{\mathbf{x}, w}) \|^2 \right. \right. \right.
$$

$$
\left. \left. \left. - \| \mathbf{x}^l - \Phi_\phi^{\mathbf{x}}(\theta_{t-1}^{\mathbf{x}, l}) \|^2 + \| \mathbf{x}^l - \Phi_{\text{ref}}^{\mathbf{x}}(\theta_{t-1}^{\mathbf{x}, l}) \|^2 \right) \right) \right]. \tag{22}
$$

Similarly, the atom types loss is designed to directly train the model output $\Phi^{\mathbf{v}}(\theta_t^{\mathbf{v}})$ to approximate the clean atom types $\mathbf{v}$, which is the $N_{\mathcal{M}} \times K$ matrix with each row as $K$-dimensional one-hot vector:

$$
\mathcal{L}^{\mathbf{v}} = \underset{\substack{(\mathbf{p}, \mathbf{v}^w, \mathbf{v}^l) \sim \mathcal{D}, t \sim [0, T], \\ \theta_{t-1}^{\mathbf{v}, w} \sim p_F(\cdot | \mathbf{v}^w, \mathbf{p}; t), \theta_{t-1}^{\mathbf{v}, l} \sim p_F(\cdot | \mathbf{v}^l, \mathbf{p}; t)}}{-\mathbb{E}} \left[ \log \sigma \left( - \beta T \left( \left( \ln p_S(\cdot | \mathbf{v}^w, \alpha_t^{\mathbf{v}}) - \ln p_R(\cdot | \Phi_\phi^{\mathbf{v}}(\theta_{t-1}^{\mathbf{v}, w})) \right) \right. \right. \right.
$$

$$
- \left( \ln p_S(\cdot | \mathbf{v}^w, \alpha_t^{\mathbf{v}}) - \ln p_R(\cdot | \Phi_{\text{ref}}^{\mathbf{v}}(\theta_{t-1}^{\mathbf{v}, w})) \right)
$$

$$
- \left( \ln p_S(\cdot | \mathbf{v}^l, \alpha_t^{\mathbf{v}}) - \ln p_R(\cdot | \Phi_\phi^{\mathbf{v}}(\theta_{t-1}^{\mathbf{v}, l})) \right)
$$

$$
\left. \left. \left. + \left( \ln p_S(\cdot | \mathbf{v}^l, \alpha_t^{\mathbf{v}}) - \ln p_R(\cdot | \Phi_{\text{ref}}^{\mathbf{v}}(\theta_{t-1}^{\mathbf{v}, l})) \right) \right) \right) \right]. \tag{23}
$$

## B. Implementaton Details

### B.1. SE-(3) Equivariant Network

To model the interaction between ligand and protein pocket atoms, we employ an SE(3)-equivariant network, PosNet3D (Guan et al., 2022), as the foundational architecture $\Phi$ in the receiver distribution. A protein-ligand graph is initially constructed by performing a $k$-nearest neighbor search on the atomic coordinates, represented as $G = \langle V, E \rangle$. At each layer, the atomic hidden states $\mathbf{h}^l$ and coordinates $\mathbf{x}^l$ are iteratively updated as follows:

$$
\mathbf{h}_i^{l+1} = \mathbf{h}_i^l + \sum_{j \in N_G(i)} \phi_h \left( d_{ij}, \mathbf{h}_i^l, \mathbf{h}_j^l, e_{ij}, t \right)
$$

$$
\Delta \mathbf{x}_i = \sum_{j \in N_G(i)} (\mathbf{x}_j^l - \mathbf{x}_i^l) \phi_x \left( d_{ij}, \mathbf{h}_i^{l+1}, \mathbf{h}_j^{l+1}, e_{ij}, t \right)
$$

$$
\mathbf{x}_i^{l+1} = \mathbf{x}_i^l + \Delta \mathbf{x}_i \cdot \mathbf{1}_{mol}
$$

where $N_G(i)$ represents the neighborhood of atom $i$ in graph $G$, while $\mathbf{h}_i, \mathbf{x}_i$ and $\mathbf{h}_j, \mathbf{x}_j$ denote the hidden states and coordinates of atoms $i$ and $j$, respectively. The term $d_{ij}^l$ represents the Euclidean distance between atoms $i$ and $j$, and $e_{ij}$ indicates whether the interaction is between protein atoms, ligand atoms, or a combination of both. The indicator function $\mathbf{1}_{mol}$ ensures that only ligand atom positions are updated. The functions $\phi_h$ and $\phi_x$ act as attention mechanisms, utilizing $\mathbf{h}_i^l$ as a query and $[\mathbf{h}_i^l, \mathbf{h}_j^l, e_{ij}]$ as keys and values.

For the initial layer, the atomic positions and hidden states are initialized as $\mathbf{x}^0 = [\boldsymbol{\mu}, \mathbf{x}_{\mathcal{P}}]$ and $\mathbf{h}^0 = \text{linear}(\theta^v, \mathbf{v}_P, t)$. In the final layer, the network produces an estimation $\hat{\mathbf{x}} = \Phi^{\mathbf{x}}$. Additionally, for the discrete variable $\mathbf{v}^{(d)}$, the function $\Phi$ applies a softmax operation to yield the probability distribution:

$$
\hat{\mathbf{v}}^{(d)} = \text{softmax} \left( (\Phi^{\mathbf{v}})^{(d)} \right).
$$

### B.2. Data Featurization

Each protein atom is characterized by multiple features to effectively capture its structural and biochemical properties. These features include a one-hot encoded element identifier (comprising H, C, N, O, S, and Se) to specify the atomic type, a one-hot encoded amino acid type indicator of 20 dimensions to classify the amino acid identity, a binary flag to distinguish backbone atoms, and a one-hot encoded arm/scaffold region indicator that designates whether the atom is part of an arm or scaffold region based on its spatial proximity to the arm prior center.

Similarly, ligand atoms are represented using a one-hot encoded element type indicator (spanning C, N, O, F, P, S, and Cl) and a one-hot encoded arm/scaffold indicator to differentiate between aromatic and non-aromatic atoms.

For efficient message passing within the protein-ligand complex, two dynamically constructed graphs are employed: a k-nearest neighbors (k-NN) graph linking ligand and protein atoms and a fully connected graph for ligand atoms. The edge features in the k-NN graph are derived from the outer products of the distance embeddings, which are computed via radial basis function expansion, and a four-dimensional one-hot encoded edge type vector that classifies different types of atomic interactions. In contrast, the ligand graph represents ligand bonds through a one-hot encoded bond type vector, which categorizes bond types as non-bonded, single, double, triple, or aromatic.

Following the existing work (Guan et al., 2023), proteins and ligands are represented through atomic coordinates alongside one-hot encoded atomic type vectors. Protein atoms are characterized by a one-hot encoded vector distinguishing among 20 amino acid types, while ligand atoms are encoded using a one-hot vector that identifies various elements, including H, C, N, O, F, P, S, and Cl. Additionally, a binary flag is incorporated to distinguish between protein and ligand atoms. To further refine these representations, two separate single-layer multi-layer perceptrons (MLPs) are applied, each mapping the input features into a 128-dimensional latent space, ensuring a compact yet informative feature embedding for subsequent computational processing.

## B.3. Preference Pair Generation

For each protein binding site, we initially generate ten molecular candidates from the pretrained model. The reward for each synthesized molecule is then evaluated based on a user-defined reward function specific to the corresponding binding site. To construct preference data, we identify a lower-performing sample with a comparatively lower reward and establish a preference ranking. The detailed procedure for reward computation and preference selection is provided in Section 4.2.

## B.4. Model Hyperparameters

For the SE(3)-equivariant network, we utilize k-nearest neighbors (kNN) graphs with a 32-nearest neighbor search to construct the graph. The model consists of nine layers, each with a hidden dimension of 128, employing a 16-head attention mechanism. ReLU activation functions are used in conjunction with Layer Normalization (Ba et al., 2016) to enhance stability and performance.

Regarding the noise schedules, we set the parameters as follows: $\beta_1 = 1.5$ for atom types and $\sigma_1 = 0.03$ for atom coordinates. The model is trained using a discrete-time loss function over 1000 training steps.

For pretraining, we adopt the same architecture as MolCRAFT, which is based on Bayesian Flow Networks (Graves et al., 2023).

For finetuning, we employ the Adam optimizer with a learning rate of 0.005 and a batch size of 32. Additionally, an exponential moving average (EMA) of model parameters is maintained with a decay factor of 0.999. Training converges within two epochs on a single RTX 4090 GPU, requiring approximately six hours for completion. For sampling, we generate 100 sample steps using a noise-reduced sampling strategy.

To ensure a higher value corresponds to a better reward across different metrics, we use a transformation function in Equation (10) as normalizing each rewards.

$$\hat{r}_i^{(j)} = \frac{\exp\big(f_j(r_i^{(j)})\big)}{\sum_{b=1}^{B} \exp\big(f_j(r_b^{(j)})\big)}$$

$$f_j(r) = \begin{cases} -r, & \text{if } j \text{ corresponds to the Vina Dock,} \\ \frac{10}{r}, & \text{if } j \text{ corresponds to strain energy,} \\ r, & \text{if } j \text{ corresponds to QED.} \end{cases}$$

where $r_i^{(j)}$ is the $j$-th reward for the $i$-th molecule, and $f_j(\cdot)$ is a simple transformation function to ensure each reward is reasonable. This maps each reward to the $(0, 1)$ interval, ensuring that distinct reward signals become comparable despite inherently different numerical ranges. We set $\gamma = 0.4$ in Equation (11)

### B.5. Computation Time

We report the time required for one epoch of pretraining and fine-tuning our model, comparing it to the AliDiff baseline trained with DPO. To measure the computation time, we use the official source code provided by the author and run the code on a single NVIDIA A6000 GPU.

*Table 4.* Time needed for training per one epoch (in seconds)

| Model | Pretraining | Finetuning only |
|---|---|---|
| AliDiff (DPO with diffusion model) | 7461.73 | 22,884 |
| Ours (DPO with BFN model) | 5695.40 | 16,354 |

This shows applying DPO to a pre-trained model takes 16,354 seconds with our method per one epoch, compared to 22,884 seconds for AliDiff, representing a 28.5% reduction. Our approach does not significantly increase overall training time; however, obtaining rewards from an external tool can be time-consuming. To address this, we preprocess and store rewards in advance, but this requires an offline DPO framework. A promising future direction for SBDD is reducing reliance on external tools, enabling a more iterative and efficient online training loop.

