# OpenReview forum: "Enhancing Ligand Validity and Affinity in Structure-Based Drug Design with Multi-Reward Optimization"
_ICML.cc/2025/Conference — ICML 2025 poster_

### Official Review · Reviewer_rt1G · 2025-03-13

**Overall Recommendation:** 3

**Summary:**

This paper proposes a multi-reward optimization framework to enhance ligand validity and binding affinity in structure-based drug design (SBDD). By integrating Direct Preference Optimization (DPO) with Bayesian Flow Networks (BFNs), the authors achieve joint optimization of multiple objectives, such as binding affinity strain energy, and QED. Experiments demonstrate that the proposed method is comparable to existing baseline models in terms of binding affinity and molecular validity, while also expanding the Pareto frontier in multi-objective optimization.

**Claims And Evidence:**

The main claims of the paper are supported by experiments and theory.

**Essential References Not Discussed:**

No

**Experimental Designs Or Analyses:**

1. Figure 3 shows the results of the molecular validity check. Compared to MolCRAFT, which serves as the backbone, the proposed method that utilizing multi-reward optimization shows limited improvement.

2. As a 3D method, and with the authors claiming to have expanded the Pareto frontier, there is a lack of intuitive presentation of the improvements. For example, compared to baselines, the docking poses of this method in the visualization are more realistic, and the Vina score is improved.

**Methods And Evaluation Criteria:**

For the formula in the left column of line 227, the paper does not explain how to set the temperature parameter to ensure that the rewards make sense.
For example, the Vina score is negative, with smaller values being better, while the strain energy is positive, with smaller values also being better. As rewards, higher values are better for both after applying softmax.

**Other Comments Or Suggestions:**

Line 377, right column, "DOP" ->"DPO"

**Other Strengths And Weaknesses:**

Strengths

See "Relation To Broader Scientific Literature"



Weaknesses

See "Methods And Evaluation Criteria", "Theoretical Claims", "Experimental Designs Or Analyses".

**Questions For Authors:**

Why choose Vina, SE, and QED instead of other metrics (such as Vina, SE, SA; Vina, SE, Clash, QED)? What difficulties arise when using multi-reward optimization?

**Relation To Broader Scientific Literature:**

The paper is closely related to existing literature. In the BFN section, it utilizes the backbone of the previous work, MolCRAFT [1]. The multi-reward part is inspired by Kim et al. (2024) [2], and the paper provides its own insights on combining DPO and multi-reward optimization with BFN.

[1] Qu Y, Qiu K, Song Y, et al. Molcraft: Structure-based drug design in continuous parameter space[J]. arXiv preprint arXiv:2404.12141, 2024.

[2] Kim K, Jeong J, An M, et al. Confidence-aware reward optimization for fine-tuning text-to-image models[J]. arXiv preprint arXiv:2404.01863, 2024.

**Theoretical Claims:**

1. In the right column of line 103, it is not specified what K represents. It seems to refer to the number of atomic features or atomic types. Moreover, it conflicts with the K in the formula in the left column of line 242, where K seems to represent the number of rewards.

2. Most of the formulas are not numbered, which affects readability.

3. The value of $\gamma$ in the formula in line 242 is not specified in the implementation details.

---

> ### Author Rebuttal · Authors · 2025-03-31
>
> ## [R3] Reviewer rt1G
>
> We sincerely appreciate your positive and constructive feedback. Below, we address the comments and questions raised. Due to the word limit, additional experimental results can be found in [anonymous pdf link](https://anonymous.4open.science/r/anonymouspdf-718B/anonymous%20rebuttal.pdf)
>
> ---
>
> **Q1) What difficulties arise when using multi-reward optimization?**
>
> Multi-reward optimization can be viewed as a multi-objective optimization, which sometimes underperforms compared to single-objective optimization due to task conflicts. Therefore, we encountered difficulties in designing a reward signal that simultaneously satisfies multiple metrics required for SBDD task.
>
> To provide more insight, we analyze the reward values of winning and losing samples using both all seven rewards and three rewards (ours) and report their average rewards. [Rebuttal Table 4](https://anonymous.4open.science/r/anonymouspdf-718B/anonymous%20rebuttal.pdf)  shows the average difference between winning and losing samples with seven rewards is much closer ($\times 10$) than that of the three rewards. We conjecture that the small difference introduces conflicts in the fine-tuning process, making it difficult for the model to optimize all properties.
>
> ---
>
> **Q2) Why choose Vina, SE, and QED instead of other metrics (such as Vina, SE, SA; Vina, SE, Clash, QED)**
>
> We first rule out using all possible metrics (as mentioned in Q1) and run single-reward experiments (Table 3) to identify unique, non-overlapping signals. We observe that optimizing one Vina metric improves all Vina metrics, so only one metric for binding affinity is sufficient. We chose QED over SA because improving QED also boosts SA, and we exclude clash due to its lack of improvement when used alone. Additionally, Vina and SE exhibit conflicting objectives, i.e., improving binding affinity (Vina) tends to compromise structural stability. This tendency is also observed in our single-reward experiment, so we include both to balance these competing objectives. Through this manual search, we determine that Vina, SE, and QED best complement each other in multi-reward optimization, as supported by our experimental results.
>
> For further information, we additionally conduct an experiment using the reward combination given in R3's question statement—Vina, SE, QED, and Clash—whose results are presented in [Rebuttal Table 5](https://anonymous.4open.science/r/anonymouspdf-718B/anonymous%20rebuttal.pdf). As shown in the table, this combination leads to performance improvements over the pretrained model across most metrics. However, it still underperforms compared to our final model choice.
>
> ---
>
> **W1) Details in temperature parameter to ensure the rewards make sense.**
>
> We sincerely appreciate the reviewer’s insightful question, which helped us clarify the formulation. In the revised manuscript, for clarity, we have revised the formula in line 227 as
>
> $ \hat{r}_{i}\^{(j)} = \frac{\exp\bigl(f_j(r_i^{(j)})\bigr)} {\sum\_{b=1}^B \exp\bigl(f_j(r_b^{(j)})\bigr)}$,
> where  $f_j(\cdot)$ is $ -r,  \quad  \frac{10}{r}, \quad r \ $ if $j$-th reward is Vina Dock, SE, and QED, respectively.
>
> This ensures a higher value corresponds to a better reward across different metrics.
>
> ---
>
> **W2) Limited Improvement in Figure 3 compared to MolCRAFT, which serves as the backbone**
>
> The results from Figure 3 are based solely on pass/fail criteria while presenting a broader range of physical plausibility (e.g., bond lengths, angles, etc). Therefore, we believe it is important to interpret Figure 3 in conjunction with Table 1 in the main paper. While the pass rate in Figure 3 shows a modest increase, the overall physical validity of our generated molecules has significantly improved, as shown in validity metrics, SE and clash. Additionally, we highlight that SOTA models such as IPDiff and AliDiff in binding affinity exhibit lower pass rates in Figure 3. This implies that these models sacrifice physical plausibility to achieve higher binding affinity while our approach improves both key properties.
>
> ---
>
> **W3) Lack of Intuitive 3D Visualization as a 3D method**
>
> We appreciate the reviewer’s valuable suggestion regarding the intuitive presentation of our contribution. We will incorporate additional visualizations in the revised manuscript, highlighting the docking poses of our generated ligands with improved binding affinity.
>
> ---
>
> **W4) $K$ in lines 103 and 242 is not specified and inconsistent.**
>
> We recognize the inconsistency in notation. As R3 mentioned, $K$ in line 103 represents the number of atom features, whereas $K$ in line 242 denotes the number of rewards. we have revised the manuscript to clarify and correct the error.
>
> ---
>
> **W5) Unnumbered formulas**
>
> To improve readability, we have numbered the equations accordingly in the revised manuscript.
>
> ---
>
> **W6) Unspecified value $\gamma$ in 242**
>
> We chose $\gamma = 0.4$. It has been included in the revised manuscript.

---

> > ### Comment · Reviewer_rt1G · 2025-04-05
> >
> > Thank you for providing additional experimental results and clarifications. This addresses my concerns. I will keep my positive score.

---

> > > ### Author Response · Authors · 2025-04-06
> > >
> > > Thank you for engagement in the discussion. We are glad to have addressed your concerns and truly appreciate your positive evaluation.

---

### Official Review · Reviewer_Tr2z · 2025-03-14

**Overall Recommendation:** 3

**Summary:**

This paper introduces a multi-reward optimization framework for structure-based drug design, integrating Bayesian Flow Networks (BFNs) with Direct Preference Optimization (DPO). The method aims to simultaneously optimize ligand binding affinity, synthetic accessibility, and conformational stability. By incorporating reward normalization and uncertainty-regularized ensembles, the model expands the Pareto frontier across multiple benchmarks, achieving better trade-offs between affinity and molecular properties.

**Claims And Evidence:**

Yes, it seems to be fine.

**Essential References Not Discussed:**

NA

**Experimental Designs Or Analyses:**

Yes, it seems to be fine.

**Methods And Evaluation Criteria:**

Yes

**Other Comments Or Suggestions:**

NA

**Other Strengths And Weaknesses:**

Strengths:

1. The integration of multi-reward optimization with BFNs addresses the limitations of single-objective approaches, balancing critical molecular attributes effectively.
2. Extensive comparisons with state-of-the-art models and ablation studies validate the necessity of multi-reward strategies, showing improvements in binding affinity and conformational stability.
3. The writing is clear and easy to follow.

Weakness:

1. Based on paper [1], which finetunes a diffusion model using DPO. The innovation of this paper is somewhat limited.
2. The model does not outperform baselines in ligand-protein clash detection, which may restrict practical applications.
3. Validation is limited to the CrossDocked dataset, raising concerns about generalizability to other protein targets or real-world drug discovery scenarios. However, this is primarily due to the inherent issues and limitations of the SBDD task at its current stage, rather than a problem with the paper itself.



[1]. Gu, et, al. Aligning target-aware molecule diffusion models with exact energy optimization. NeurIPS 2024.

**Questions For Authors:**

1. I'm curious about what results would be obtained if the fine-tuned model were sampled again and the fine-tuning process were repeated. Will the model continue to improve, or will its performance deteriorate due to certain reasons (such as overfitting)?

**Relation To Broader Scientific Literature:**

Refer to the strengths and weakness.

**Theoretical Claims:**

Yes, it seems to be fine.

---

> ### Author Rebuttal · Authors · 2025-03-31
>
> We deeply appreciate your thoughtful comments and positive feedback. Here, we address the comments and questions mentioned.
>
> ---
>
> **W1) The model does not outperform baselines in ligand-protein clash detection, which may restrict practical applications.**
>
> We acknowledge the reviewer’s concern regarding the clash detection performance. However, we would like to clarify the following points:
> The clash metric reflects the number of colliding atoms, which inherently depends on the molecular size. As shown in the average size of main table 1 in our paper, Pocket2Mol and AR tend to generate smaller molecules on average compared to other models, which naturally results in fewer clashes. To address this discrepancy, we conduct an additional analysis where we filter the generated ligands based on the number of atoms to obtain comparable molecular sizes and report the clash performance accordingly.
>
> Rebuttal Table 2. Clash performance after filtering for similar molecule sizes.
> |   Method  |  AR  | Pocket2Mol | TargetDiff | DecoompDiff | DecompOpt | IPDiff | AliDiff | MolCRAFT | Ours |
> |:---------:|----:|----------:|----------:|-----------:|---------:|------:|-------:|--------:|----:|
> | Avg. size | 17.7 |       17.7 |       18.0 |        17.7 |      17.2 |   17.5 |    17.6 |     17.8 | 17.6 |
> |   Clash (↓) | **4.46** |       6.24 |       7.50 |        6.70 |      9.10 |   6.20 |    7.20 |     5.87 |  4.50 |
>
>
> In Rebuttal Table 2, when comparing models with comparable molecular sizes, our method demonstrates strong performance in clash metric. Notably, the only model exhibiting a better clash score, AR, performs poorly in binding affinity. Given that both low clash and high binding affinity are critical for practical applications, these results indicate that our model offers the best balance, making it the most suitable choice for real-world deployment.
>
> ---
>
> **W2) The innovation of this paper is somewhat limited.**
>
> We acknowledge that each methodological component may not be significantly novel. However, integrating them into a single system that simultaneously enhances key properties for Structure-Based Drug Design (SBDD) remains a meaningful contribution. In particular, we are the first to apply DPO to Bayesian Flow Networks (BFNs) and to leverage reward normalization for multi-reward optimization—improving binding affinity, validity, and drug-likeness at once. This approach, not previously explored in the literature, helps make SBDD more practical.
>
> ---
>
> **W3) Validation is limited to the CrossDocked dataset, raising concerns about generalizability to other protein targets or real-world drug discovery scenarios. However, this is primarily due to the inherent issues and limitations of the SBDD task at its current stage, rather than a problem with the paper itself.**
>
> We completely agree with this point. As research in SBDD continues to grow, we expect more robust and diverse benchmarks to emerge, allowing broader validation and ensuring greater generalizability to real-world scenarios.
>
> ---
>
> **Q1) I'm curious about what results would be obtained if the fine-tuned model were sampled again and the fine-tuning process were repeated. Will the model continue to improve, or will its performance deteriorate due to certain reasons (such as overfitting)?**
>
> We appreciate the insightful question, which encourages further discussion. To address it, we provide results for a model fine-tuned iteratively for two stages (i.e., multistage) with newly sampled molecules using DPO, compared to our model and the pretrained model.
>
> Rebuttal Table 3. Performance comparison of the pretrained model, a multistage fine-tuned model, and our model,
> | Method |  Vina Score Med. (↓) |  SE Med. (↓) |   Clash Avg. (↓)  |  SA (↑)  |   QED (↑) |
> |:--------------|---------:|---------:|-----------:|---------:|------------:|
> |    Pretrained    |  -7.04 |7.62 |7.09 |0.69 |0.50|
> |     Multistage DPO (2 stage)    |  -7.16 |  **5.15** |   7.54 |  **0.74**|   0.54  |
> | Ours |  **-7.38** |  5.56   |  **6.69** | **0.74**  | **0.55**|
>
> Although the two-stage DPO achieves competitive performance, it does not surpass our model choice. Xu et al. also suggest that fine-tuning via DPO does not necessarily guarantee significant improvements [1]. Furthermore, iteratively measuring rewards using external tools at each training step is computationally expensive compared to the training process itself, which led us to adopt an offline strategy. Nevertheless, we believe that exploring an online, iterative approach with efficient sampling remains a promising direction for future research.
>
>  **Reference**
>
> [1] Xu, Shusheng, et al. "Is dpo superior to ppo for llm alignment? a comprehensive study." arXiv preprint arXiv:2404.10719 (2024).

---

> > ### Comment · Reviewer_Tr2z · 2025-04-08
> >
> > Thank you for your response, it has basically resolved my concern, and I will maintain a positive score.

---

> > > ### Author Response · Authors · 2025-04-09
> > >
> > > We appreciate the constructive discussion and continued positive score. We are glad we can address the concerns raised

---

### Official Review · Reviewer_e8Ki · 2025-03-18

**Overall Recommendation:** 3

**Summary:**

This paper introduces a multi-reward optimization framework for structure-based drug design, addressing the challenge of generating ligand molecules with multiple desired properties like binding affinity, validity, and drug-likeness.  It fine-tunes generative models for these attributes together, using direct preference optimization for a Bayesian flow network and a reward normalization scheme. Experimental results show the method generates more realistic ligands with higher binding affinity compared to baselines, expanding the Pareto front observed in previous studies.

**Claims And Evidence:**

The main claim of this work is that the proposed multi-reward optimization framework generates more realistic ligands than baseline models. While not all evaluation metrics outperform baselines, the authors argue that their approach expands the Pareto front.

**Essential References Not Discussed:**

N/A

**Experimental Designs Or Analyses:**

- Lack computation complexity analysis against diffusion baselines. might be complex given the complexity of Bayesian Flow Networks and multi-reward optimization

**Methods And Evaluation Criteria:**

- Method: The paper applies multi-reward optimization and Direct Preference Optimization (DPO) for ligand generation in structure-based drug design. While the approach aligns with recent advances, it closely resembles existing frameworks (Kim et al., 2024) and apply the method on conditional generation of SBDD.

- Evaluation: The evaluation criteria are well-structured, incorporating binding affinity, synthetic accessibility, strain energy, and drug-likeness metrics. The use of benchmark datasets like Cross-Docked ensures comparability with prior work.

**Other Comments Or Suggestions:**

N/A

**Other Strengths And Weaknesses:**

Minor issue:
- figure 1&2 should provide axis indications

**Questions For Authors:**

N/A

**Relation To Broader Scientific Literature:**

- Structure-based Drug Design: A task to generate ligand conditional on protein targets. Previous works explore from auto-regressive models [GraphBP, Pocket2mol, etc] to diffusion-based models [TargetDiff, IPDiff, DecompDiff etc] and optimal transport based models [DecompOpt etc].

- Bayesian Flow Networks [Graves et al., 2023].

- Direct Preference Optimization [DPO 2024], and the multi-reward design in this work follows [Kims 2024].

**Theoretical Claims:**

The derivation of the loss function for Eq. 1 in Appendix A appears correct, though I did not examine it in detail.

---

> ### Author Rebuttal · Authors · 2025-03-31
>
> We deeply appreciate your efforts and positive feedback. Here, we address the comments and questions mentioned.
>
> ---
>
> **W1)  Lack of computation complexity analysis against diffusion baselines. might be complex given the complexity of Bayesian Flow Networks (BFNs) and multi-reward optimization**
>
> To address your concern regarding time complexity, we report the time required for one epoch of pretraining and fine-tuning our model, comparing it to the AliDiff baseline (which also uses DPO). To measure the computation time, we use the official source code provided by the author and run the code on a single NVIDIA A6000 GPU.
>
> Rebuttal Table 1.  Time needed for training per one epoch (in seconds)
> | Model                     | Pretraining  | Finetuning only |
> |:---------------------------:|:----------------------------:|:-----------------:|
> | AliDiff (DPO with diffusion model) | 7461.73                  | 22,884      |
> | Ours (DPO with BFNs)         | 5695.40                    | 16,354       |
>
> The Rebuttal Table 1 shows applying DPO to a pre-trained model takes 16,354 seconds with our method per one epoch, compared to 22,884 seconds for AliDiff, representing a 28.5% reduction. These results demonstrate that our model is more time-efficient than the diffusion-based baseline. Furthermore, coupled with its strong performance, this highlights the practicality of our model for Structure-Based Drug Design (SBDD) tasks. We have added this computation time in the appendix and appreciate R1's feedback in helping us improve the manuscript.
>
> Our approach does not significantly increase overall training time; however, obtaining rewards from an external tool can be time-consuming. To address this, we preprocess and store rewards in advance, but this requires an offline DPO framework. A promising future direction for SBDD is reducing reliance on external tools, enabling a more iterative and efficient online training loop.
>
> ---
>
> **W2) Minor issue: figure 1&2 should provide axis indications**
>
> Thank you for the editorial comment. We have added the axis indications in our revised manuscript.

---

> > ### Comment · Reviewer_e8Ki · 2025-04-07
> >
> > I thank the authors for their rebuttal. I believe my concerns have been generally addressed, and I keep my decision of weak acceptance.

---

> > > ### Author Response · Authors · 2025-04-07
> > >
> > > We appreciate the positive recommendation and the constructive feedback provided during the discussion. All comments help further refine the paper.

---

### Decision · Program_Chairs · 2025-05-01

**Decision:**

Accept (poster)

**Comment:**

This paper introduces a multi-reward optimization framework based on DPO for structure-based drug design. It aims to optimize multiple desired properties like binding affinity, validity, and drug-likeness.  Experimental results show the method generates more realistic ligands with higher binding affinity compared to baselines and a better Pareto front.

Strengths:
- The integration of multi-reward optimization with BFNs addresses the limitations of single-objective approaches, balancing critical molecular attributes effectively.
- Experiments show improvement in binding affinity and conformational stability.

Weakness:
- The paper uses existing training approach for multi-objective ligand optimization. The novelty is limited.
- Validation is on the CrossDocked dataset, which is synthetic.

Other concerns raised by reviewers such as computational complexity,  ligand-protein clash results, choice of metrics, and temperature parameter are addressed by authors during rebuttal.